# Exploring Factors Associated with Physical Activity in the Elderly: A Cross-Sectional Study during the COVID-19 Pandemic

**DOI:** 10.3390/bs14010062

**Published:** 2024-01-17

**Authors:** Vesna Miljanovic Damjanovic, Lejla Obradovic Salcin, Daria Ostojic, Ljerka Ostojic, Barbara Gilic, Marijana Geets Kesic, Edin Uzicanin, Damir Sekulic

**Affiliations:** 1Clinic for Physical Medicine and Rehabilitation, University Hospital Mostar, 88000 Mostar, Bosnia and Herzegovina; vesna.m.damjanovic@fzs.sum.ba (V.M.D.); lejla.o.salcin@fzs.sum.ba (L.O.S.); daria.ostojic@mef.sum.ba (D.O.); 2Faculty of Health Sciences, University of Mostar, 88000 Mostar, Bosnia and Herzegovina; 3Faculty of Medicine, University of Mostar, 88000 Mostar, Bosnia and Herzegovina; 4Academy of Sciences and Arts of Bosnia and Herzegovina, 71000 Sarajevo, Bosnia and Herzegovina; ljerka.ostojic@kifst.eu; 5Faculty of Kinesiology, University of Split, 21000 Split, Croatia; barbara.gilic@kifst.eu (B.G.); markes@kifst.hr (M.G.K.); 6Faculty of Sport and Physical Education, University of Tuzla, 75000 Tuzla, Bosnia and Herzegovina; edin.uzicanin@untz.ba

**Keywords:** physical literacy, physical activity, lock down, residence status, lifestyle medicine

## Abstract

The COVID-19 pandemic negatively influenced individuals’ physical activity levels (PALs) and particularly the PAL of the elderly. However, few studies have examined the correlates of PALs in this population during the pandemic. This study aimed to evaluate the residence-specific correlates of PALs in elderly people from Croatia and Bosnia and Herzegovina during the COVID-19 pandemic. The participants were 211 persons older than 65 years (101 females), of whom 111 were community-dwelling residents, and 110 were nursing home residents (71.11 ± 3.11 and 72.22 ± 4.01 years of age, respectively; *t*-test = 0.91, *p* < 0.05). The variables included health status, residential status sociodemographic factors, anthropometrics (body mass, height, and body mass index), and PAL. PAL was evaluated using a translated version of the Physical Activity Scale for the Elderly (PASE), and was validated in this study. PASE showed good test–retest reliability (51% of the common variance) and validity (57% of the common variance, with the step count measured using pedometers). Apart from participants’ health status and age, PAL was positively correlated with (i) community-dwelling residence (OR = 1.93, 95% CI: 1.60–2.23), and (ii) a lower BMI (OR = 0.85, 95% CI = 0.71–0.98). The pre-pandemic physical activity was positively correlated with the PAL of the nursing home residents (OR = 1.2, 95% CI: 1.02–1.45). A higher education level was positively correlated with the PAL of community-dwelling residents (OR = 1.31, 95% CI: 1.04–1.66). This study evidenced the residence-specific correlates of PALs, and enabled the identification of specific groups that are at risk of having low PALs during the pandemic. Future studies examining this problem during a non-pandemic period are warranted.

## 1. Introduction

Regular physical activity (PA) is one of the most important pillars of lifestyle medicine, and aids in preventing chronic diseases and maintaining general health [1]. Indeed, physical inactivity is among the four leading causes of all-cause mortality; all-cause mortality involves increased morbidity and mortality arising from “lifestyle diseases” such as obesity, type II diabetes, cardiovascular diseases, and some types of cancers [2,3]. Thus, maintaining adequate physical activity levels (PALs) is considered to decrease the risk of premature death by delaying aging [4]. Older adults and the elderly are at an increased risk of developing chronic diseases due to decreased PALs [5]. It is a well-known fact that individuals’ PALs decrease with aging [6]. A study on more than 92,000 participants reported that participation in physical exercise declines throughout adult life, with only half of adults and only a quarter of people aged more than 65 years meeting the minimum recommended PALs [7]. 

One of the important aspects of health status in the elderly is housing [8]. This is particularly the case considering the differences in PAL and factors associated with PAL between older people living in their own homes (e.g., community-dwelling people), and those living in nursing homes (institutions) [9,10]. Specifically, in care institutions, physical activity is mostly organized by the nursing home, while barriers to physical activity include personal factors such as health problems and fear of injury, as well as environmental factors like lack of understanding and restrictions [11,12]. On the other hand, among older people living at home, customary physical activity is generally low and self-organized, with age, health status and sex being key determinants [13]. 

The pandemic of the SARS-CoV-2 virus (i.e., the COVID-19 pandemic) resulted in drastic changes in social and lifestyle behavior [14,15]. Specifically, social distancing measures were introduced in 2020 to prevent the rapid spread of the virus; these included banning social gatherings, closing schools and universities, closing sports facilities, and even limiting permissions to go outside for groceries and similar “normal” life behaviors [16]. As a consequence of these social distancing measures, numerous studies reported a significant change in the PALs of different populations, including children, adolescents, older adults, and the elderly (i.e., the global population) [17,18,19,20]. It is noteworthy that the elderly population is at an increased risk of experiencing the serious consequences of reduced PALs, as this can lead to a worsening of their comorbidities. Briefly, reduced PALs during the pandemic might have led to reductions in musculoskeletal strength and endurance, and in cardiorespiratory capacity, leading to a loss of muscle function and motor control, and the development of sarcopenia and cardiometabolic disorders [20,21]. Altogether, this problem is even more evident if we consider that older adults and the elderly make up the majority (>80%) of deaths caused by the SARS-CoV-2 virus [22]. 

Not surprisingly, several previous studies have examined the PALs among the elderly during the COVID-19 pandemic. Specifically, a study on individuals aged over 60 years from Italy noted a reduction in the total weekly PA energy, which was confirmed in elderly Spanish individuals aged 82.4 ± 6.1 years [23,24]. Moreover, a study on elderly French individuals aged 69.7 ± 4.2 years reported that PALs decreased for 39.2% of people during the lockdown; meanwhile, a study on Chinese individuals aged more than 60 years recorded that the prevalence of insufficient PALs doubled during the initial period of the COVID-19 pandemic [25,26]. Collectively, reduced PALs in the elderly population were recorded at a global level. At this point, it has to be emphasized that this noted reduction in PALs is associated with sarcopenia, increased frailty, and a worse self-perception of health, which may have led to irreversible consequences during this emergency period [27]. Therefore, the factors that might have influenced the PALs of elderly individuals during the COVID-19 pandemic should be determined and pointed out, in an attempt to prevent and slow down the negative health consequences of reduced physical movement. 

The strictest lockdowns and social distancing measures were imposed in China, Italy, and Spain, and thus a significant reduction in PALs has been evidenced in these countries. Such changes in PAL have also been evidenced in the territories of Southeastern Europe [16,25,28]. However, the majority of studies investigating PALs and the correlates of PALs in the pandemic period in Southeastern Europe have mostly focused on children and adolescents; meanwhile, studies investigating PALs in the elderly are lacking. Considering the specifics of this region, the investigation of PAL in elderly would be particularly important. In brief, older people living in this region experienced wars in the early 1990s, which dramatically changed their lives, and in some cases places of residence (because of the war acts, large migrations of ethnic groups in the territory of the former Yugoslavia occurred in the 1990s). Also, during the past 10 years, new European Union (EU) members from the region faced economic migrations to western EU countries, which consequently resulted in a lack of familial support for elderly people [29]. 

Therefore, the aim of this research was to investigate the anthropometric-, sociodemographic-, and health-related-factors associated with PALs in elderly participants during the pandemic period, considering their place of residence (community-dwelling and nursing home residents). Initially, we hypothesized that the studied factors would be differentially associated with PALs during the pandemic period in community-dwelling and nursing home residents. Additionally, we evaluated the reliability and validity of the Physical Activity Scale for the Elderly (PASE), translated into the Croatian/Bosnian and Herzegovinian language. 

## 2. Materials and Methods

### 2.1. Participants and Study Design

Participants in this study were persons older than 65 years (+65 years) from Bosnia and Herzegovina, and Croatia. The total sample comprised 211 participants (101 females), of whom 111 were community-dwelling residents (age: 71.11 ± 3.11 years) and 110 were nursing home residents (72.22 ± 4.01 years of age; *t*-test = 0.91, *p* > 0.05). Participants were contacted directly by their personal medical doctor 5–6 months before the initiation of the study (December 2021) and were asked to participate in the study, which was planned for spring 2022. The main part of the study was performed during the COVID-19 pandemic, when certain prevention and social distancing measures remained in place, although there was no strict lockdown in the studied countries. Specifically, face masks were mandatory in all health institutions (including nursing homes), and public transport, nursing home visits were limited, and visitors were health-checked, while social distancing of 1.5 m in public places was mandatory. Altogether, 360 invitations were sent, and 224 of the invited individuals responded positively. The final sample (*n* = 211) did not include those who died in the meantime (four persons) and/or changed their place of residence (six persons). A subsample of 61 participants tested in the main part of the study was included in the validity study, which was completed during November and December 2022 (please see below for details). The study was conducted in accordance with the Helsinki declaration and was approved by the Ethical Committee of the Faculty of Kinesiology, University of Split, Croatia. The study design and testing are presented in Figure 1. 

### 2.2. Variables and Measurement 

The variables in this study included participants’ age (in years), anthropometrics (body mass and height, and calculated body mass index—BMI), residential status (community-dwelling and nursing home residents), PALs, and sociodemographic- and health-related variables. After obtaining consent for participation in the study (please see above for details), anthropometric data were collected from the participants’ medical record by the family physician. Other data were collected via phone interviews. 

The participants’ PALs were evaluated using the Physical Activity Scale for the Elderly (PASE). Originally, the PASE was constructed to assess self-reported PALs in individuals over 65 years of age [30]. It consists of 12 items/questions about occupational, household, and leisure time activities during the previous week (7-day recall period). It assesses typical activities for this population, including walking, exercise, yard work, and housework. It uses the duration, frequency, and intensity of the activity to assign a score ranging from 8 to 793 (a higher score means higher PALs). The questions are scored differently; involvement in leisure time and strengthening activities is scored as 1 (never), 2 (1–2 per week), 3 (sometimes), and 4 (often), while the duration of both activities is scored as 1 (less than 1 h), 2 (1–2 h), 3 (2–4 h), and 4 (more than 4 h). Work-related and household activities are scored as yes or no [31]. Because no study has so far validated the PASE in the Croatian/Bosnian and Herzegovinian language (local language), for the purpose of this investigation, the PASE was translated into the local language by two independent translators (both medical doctors with high academic expertise in the English language); after back-translation, the PASE was used in this investigation (Appendix A). Apart from raw scores, PASE results were dichotomized (low PAL coded as “1” (lower 50 percentiles) vs. high PAL coded as “2” (higher 50 percentiles) for logistic regression calculations (see Section 2.3 for details). 

When they were tested using the PASE during the pandemic period, all participants responded to questions about their level of education (no school, elementary school, high school, or college/university level), self-determined pre-pandemic PALs (low level (mostly sedentary), below average level (only basic at home), average PALs (regular home duties and sporadic walking 1–2 times per week), above average PALs (regular home duties, everyday walking >500 m), or high PALs (regular home duties, everyday walking >500 m, active transportation and/or mild physical work at least twice per week)), the usage of the walking aids (yes or no), and participation in any organized physical activity in the pre-pandemic period (yes/no). On the same occasion, the participants’ health status was checked using the Short Form Health Survey (SF-12), which is a patient-reported survey of health [32]. In brief, the SF-12 is a tool that aims to evaluate individuals’ physical (SF-12_PCS) and mental health (SF-12_MCS), and is regularly used in order to test individuals’ overall health status in a reliable, meaningful, and non-time-consuming way; as such, it has been used by [33,34]. The interviews were conducted by experienced interviewers, the authors of this paper. While practically all questions were standardized, we performed structured interviews, while only in some cases additional explanations were needed (i.e., when a participant did not recognize the question, and the interviewer had to present some type of physical activity or health condition in more figurative speech). The ordering and wording of the questions were similar for all interviewees, and a pre-defined specific protocol was used during interviews (i.e., personal presentation of the interviewer, introduction, main part of interviewing, conclusion). Such a protocol minimized the potential biases and the required time, and the conductor controlled the process. 

For the purpose of the validity study, a convenient subsample of 61 participants (of whom 28 were community-dwelling and 34 were females) responded to the questionnaire twice (test–retest procedure) in a time frame of 10–14 days, which allowed us to evaluate the reliability of the translated tool. In the period between the test and retest, the physical activity of the participants involved in the validity study was measured using pedometers (Yamax CW-701 P) over a period of five days, and the average number of steps for one day was later used as an objective measure of the PALs of the studied participants. 

### 2.3. Statistics

After checking the normality of the distributions using the Kolmogorov–Smirnov test, the descriptive statistics included means and standard deviations (for normally distributed variables), and frequencies and percentages (for nonparametric data). 

The reliability analysis for the PASE included the calculation of the test–retest reliability coefficients as previously suggested [30], but for additional analysis, we also calculated the percentage of equally responded queries (i.e., total test–retest agreement). The validity of the PASE was checked via Spearman’s correlation between the PASE score and the step counts obtained using the pedometer. 

The association between the studied sociodemographic, health-related, and anthropometric indices (predictors) and binarized PAL (criterion) was checked by calculating a series of logistic regressions. Previous studies evidenced that the primary limiting factor for PALs is a poor health status [35], which is additionally and logically correlated with age. Therefore, in the preliminary statistical processing, we checked the correlations between the age and health status (evaluated using SF-12_PCS, and SF-12_MCS) and the PALs of our participants. Since the correlations were strong (please see Results for details), the participants’ age and health status were included as covariates in all subsequent logistic regression calculations. In the first phase, we checked associations between all predictors and PALs. Since the analysis on the total sample evidenced a significant association between the participants’ residential status and PALs (please see below for details), in the second phase, all logistic regressions were stratified according to residential status (e.g., calculated separately for community-dwelling and nursing home residents). The odds ratios (ORs) with corresponding 95% confidence intervals (95% CIs) were reported, and Statistic aver. 13.5 (Tibco Inc., Palo Alto, CA, USA) was used for all statistical analyses.

## 3. Results

Table 1 presents the descriptive statistics for the study variables in the total sample, separately for males and females, with significant *t*-test differences between genders. In general, males were taller, and females had a higher BMI (*p* < 0.05). 

### 3.1. Descriptive Statistics 

Descriptive statistics and differences between genders are presented in Table 1

### 3.2. Reliability and Validity

The test–retest reliability of the PASE was appropriate, with a correlation of 0.71 (Figure 2). 

The percentage of absolute agreement between the test and retest for the PASE is presented in Appendix A. In brief, apart from the “open-ended questions”, identical responses were least common for question 2a: “Over the past 7 days, how often did you take a walk outside your home or yard for any reason? For example, for fun or exercise, walking to work, walking the dog, etc./On average, how many hours per day did you spend walking?”. In total, 54% gave identical responses. The highest agreement between the test and retest was evidenced for the following questions: “During the past 7 days, have you done any light housework, such as dusting or washing dishes?” and “During the past 7 days, have you done any heavy housework or chores, such as vacuuming, scrubbing floors, washing windows, or carrying wood?” (75% and 76% of participants, respectively, responded identically).

The validity of the PASE, regarding its correlation with the number of steps measured using pedometers, is presented in Figure 3. The correlation between the objectively measured PALs obtained using pedometers and the PASE showed that these two measurements share 57% of the common variance when observed for the total sample (i.e., when not separating community-dwelling and nursing home residents) 

The correlation between the objectively measured PALs and the PASE was evidently lower in nursing home residents (43% of the common variance) (Figure 4A) than in community-dwelling residents (69% of the common variance) (Figure 4B). Also, the correlation was somewhat weaker for females (50% of the common variance) (Figure 4C) than for males (64% of the common variance) (Figure 4D). 

The correlation analyses showed a strong correlation between the SF-12 scores and the PASE (Pearson’s R = 0.81 and 0.86 for SF-12_MCS and SF-12_PCS, and PASE, respectively; *p* < 0.001), showing the strong association between PALs during the pandemic period and the generic patient-reported measure of health status. Also, the PALs obtained using the PASE were significantly correlated with the participants’ age (Pearson’s R = 0.71; *p* < 0.001). Therefore, when further calculating the logistic regressions, the SF-12_MCS, SF-12_PCS, and participants’ age were systematically controlled for confounding effects (e.g., included as covariates). 

### 3.3. Correlates of Physical Activity Levels

The results of the logistic regressions for the studied criteria, with the SF-12_MCS, SF-12_PCS, and participants’ age being used as confounding factors for the total sample of participants, are presented in Figure 5. The variables significantly correlated with the PASE score during the COVID-19 pandemic were the participants’ residence status and BMI. More specifically, being a community-dwelling resident increased the participants’ likelihood of being more physically active during the pandemic, at 93% (OR = 1.93, 95% CI: 1.60–2.23). Also, a higher BMI decreased the likelihood of having higher PALs (OR = 0.85, 95% CI: 0.71–0.98).

When calculated separately for community-dwelling participants, the significant correlate of PALs obtained using the PASE was the participants’ educational level, with a higher pandemic PAL found in those who had a higher education level (OR = 1.31, 95% CI: 1.04–1.66) (Figure 6A). Among nursing home residents, the participants’ pandemic PALs were positively correlated with their pre-pandemic PALs (OR = 1.2, 95% CI: 1.02–1.45) (Figure 6B). 

## 4. Discussion

The aim of this research was to investigate the factors associated with PAL during the COVID-19 pandemic in the elderly. In addition, we evaluated the psychometric properties of the translated version of the PASE. Accordingly, the main findings were as follows: (i) the translated version of the PASE showed appropriate reliability and validity in the studied participants, with some differences in validity for different subgroups of participants; (ii) apart from the participants’ health status and age, the significant correlates of PALs during the COVID-19 pandemic were their residential status and BMI; and (iii) when looking specifically at nursing home residents, the participants’ PALs during the pandemic were correlated with their pre-pandemic physical activity, while a higher education level was correlated with better PALs in community-dwelling residents. Considering the differential correlates of PALs in residence-specific groups of participants, our initial study hypothesis can be accepted.

### 4.1. Psychometric Properties of the Physical Activity Questionnaire for Elderly 

Since the PASE has not previously been validated in southeastern Europe (a territory of former Yugoslavia where similar languages are spoken), in this study, we evaluated the reliability and validity of the PASE translated into the Croatian/Bosnian and Herzegovinian language (it should be noted that these two languages are very similar and the translated version of the PASE was absolutely understandable to participants from both countries). The reliability of the PASE was checked using the test–retest procedure on a convenient sample of participants; this procedure is generally accepted when studying reliability for similar questionnaires [36,37]. Generally, the results showed the proper stability of the measurement, especially considering proper reliability indices for some highly sensitive questions, which are clearly influenced by non-systematic differences in life circumstances that could occur between testing sequences, such as differences in weather conditions. 

We must note that the reliability (and validity) of the PASE was checked in the specific period, namely, during the COVID-19 pandemic. This is important to highlight since the stability of the responses during test and retest is of the utmost importance for the proper reliability of any measurement tool [38]. In the study period, the life habits of the studied participants were relatively stable, which increased the likelihood of identical responses given during test and retest, and could have influenced the high reliability of the PASE. Despite this, the reliability of the PASE was found to be in accord with previous research that confirmed the stability of the same instrument in elderly Japanese, Turkish, and Chinese individuals [39,40,41]. 

As presented, the validity of the PASE was appropriate for all participants, although it seems that the correlation between the PASE and pedometer-measured PALs was higher in community-dwelling residents. This is explainable with regard to the structure of the PASE itself, and differences between the types of PALs: (i) between males and females, and (ii) between participants living in nursing homes and community-dwelling residents. 

With regard to gender differences in validity, we can logically assume that the greatest part of the PALs in male participants is related to walking, and therefore the step count measured using pedometers actually represents the PALs among > 65-year-old males relatively accurately. In the meantime, pedometers themselves are probably less accurate for the evaluation of the PALs among females, whose PALs are known to be more influenced by home duties (i.e., cleaning, cooking, or ironing), which are not adequately represented by step count [42,43]. As a result, we may suggest that, although the correlation between the PASE and step count was lower in females, the ecological validity of the PASE could be even better for females than for males. 

A similar explanation is probable even for the higher correlation between the PASE and step count in community-dwelling participants than in nursing home residents. In brief, community-dwelling participants were more likely to take a higher number of steps per day simply because of their regular home duties. They individually went to shops and pharmacies, prepared the food, did the cleaning, etc., which altogether increase the number of steps. Consequently, their PAL is directly influenced by “step count”. Meanwhile, this is not as likely for nursing home residents, whose PALs are mostly associated with organized recreation in nursing homes, and in most cases are not based on the (higher) number of steps, but are based on standing- and sitting-exercises. Therefore, simply because of this issue, the validity of the PASE in this study sample was expected to be somewhat lower in nursing-home residents. However, as in the previously discussed gender specifics, this difference in validity should not be considered crucial. In conclusion, we can highlight that our results are in line with previous research, in which the validity of the PASE was confirmed in elderly individuals from the US and Japan [31,39]. 

### 4.2. Predictors of Physical Activity Levels during the COVID-19 Pandemic

The strongest predictors of PALs during the COVID-19 pandemic were health status and age, with higher PALs found in those with higher scores in the SF-12 (better health status) and younger age. The negative correlation between age and PALs is well documented, and explained by numerous factors including lower functional fitness, a higher prevalence of depression, and different psychosocial factors (i.e., lack family support, a poor awareness of exercise benefits, and expectations) [44,45]. Most probably, these explanations are also plausible in our study. However, we will briefly discuss the association between health status and PALs. 

There is no doubt that being healthy enabled the participants to be physically active even during the pandemic. Specifically, people with good health conditions were more likely to have the capacity to participate in any form of PA, and such a relationship is directly and indirectly confirmed in numerous studies [46,47]. For example, PA has been associated with individuals’ quality of life and physical health status [47]; a whole intervention study that investigated the effects of PA intervention on people aged 66.61 ± 4.73 years noted an improved quality of life and health status in the group that underwent the PA intervention compared to the control group [48]. Hence, while a good health status could “allow” participation in physically demanding activities, it was also expected that participating in any form of PA would result in a good health status in the elderly population studied here. However, for the purpose of this study, it is also important to highlight the possibility that those respondents who had good health were physically active before the pandemic and tried to maintain their PALs even during the crisis period (i.e., the COVID-19 lockdown); this is indirectly supported by the association between the pre-pandemic PALs and the PASE during the pandemic (please see below for more details).

Maintaining PALs in older age helps with healthy aging, and the concept of “physical literacy” (PL) is thought to be the cornerstone of maintaining PALs throughout life [49]. To be precise, PL is considered to be a lifelong journey of ongoing commitment to participating in PA [50]. What is important to note is that physically literate adults and elderly individuals are capable of adapting their PA routines when faced with challenges such as injury, illness, or crisis situations [51]. Also, elderly individuals with developed PL skills can remain independent, which is associated with healthy aging [52]. Further, healthy aging is associated with modifying activities and remaining physically active, which enables elderly individuals to maintain high levels of functioning across several dimensions of life [53]. Collectively, maintaining and developing PL optimizes opportunities for a good health status across all stages of life, and this could be the reason why we obtained such results. 

Another factor correlated with the PASE was BMI, with lower PALs found in those with a higher BMI. This is not surprising, considering that numerous studies have reported associations between BMI, obesity/overweight, and PALs [54,55,56]. In particular, a study on individuals aged 67 ± 6.2 years found that the total leisure time spent undertaking PA was inversely associated with BMI [56]. Moreover, a study on adults from 10 countries recorded that BMI and the odds of being overweight decreased linearly with an increase in daily PALs (to be precise, an increase in moderate-to-vigorous PA from 0 to 40 min a day) [55]. Also, the same study investigated the moderating effects of gender on the associations between PALs and BMI, and reported mixed findings; in some countries, the associations were stronger in men, while the associations were stronger in women in other countries, which means that the region plays a role in these associations [55]. One could argue that our results of a negative correlation between BMI and PALs are not in agreement with the known fact that a higher BMI in elderly individuals is associated with a lower mortality risk [57]; this is certainly true, but in our case, not so relevant, mostly because our participants had regular (and not low) BMI values. 

The predictor of PALs during the pandemic was residential status, with higher PALs in community-dwelling residents. This is understandable considering that the living conditions for older people are completely different if they live in nursing homes or in their homes. Previous studies provided insights into the differences in the PALs between community-dwelling and nursing home residents. For example, Brach et al. (2019) found that older adults residing in planned group residential settings, which provide supportive services, were more sedentary than those residing in private homes [58]. In a very recent study, Siltanen (2023) compared senior housing residents to community-dwelling older adults and found that men in senior houses had lower levels of active aging compared to community-dwelling men, while women in senior houses had a greater desire to be active but faced poorer opportunities for activity compared to community-dwelling women [59]. 

There is no doubt that nursing homes offer residents the care of qualified professionals, which also includes help with engagement in any form of PA [60]. A study that investigated PA in German nursing homes noted that a variety of exercise types and forms were available in almost all of the included nursing homes (i.e., more than six different forms of exercise offered to residents) [61], and we can see that the situation is similar in the nursing homes observed in this study. However, during the pandemic period, due to limitations on social gatherings, such activities in nursing homes were limited, and were organized only when weather conditions allowed exercising in open spaces. Not surprisingly, community-dwelling residents had more opportunities for physical activity during the pandemic than nursing home residents. First, community-dwelling residents had to complete regular home duties (which was not the case for nursing home residents), and second, even if they had someone to help them with home duties during the pre-pandemic period, this help was limited during the pandemic due to social distancing [62].

In other words, it is almost certain that the measures imposed during the pandemic differentially influenced the PALs of community-dwelling and nursing home residents.

When looking at only those who live in nursing homes, the strongest predictor of PA during the pandemic are the individuals’ pre-pandemic PALs. These results can be interpreted in several ways. Firstly, numerous studies have confirmed the trend observed in PALs throughout life [63,64]. In particular, being active at baseline enhanced the likelihood of being active at the 20-year follow-up point, according to a 20-year tracking research study that prospectively observed participants aged 48.6 ± 5.4 years at baseline [63]. Secondly, the finding that previous activity was the strongest predictor of PA during the COVID-19 pandemic can be again discussed in terms of PL. To be precise, people with better PL know the importance of staying physically active and thus try to maintain their PALs [50]. PA has numerous health benefits, especially for older adults and the elderly, such as the prevention of falls, decreased frailty and bone damage, improved cardiorespiratory functions, and an improved general quality of life [65]. As a result, those who engaged in physical activity in the pre-pandemic period were probably more proficient in PL, which encouraged them to continue being active and inspired them to engage in physical exercise throughout the pandemic.

Among community-dwelling participants, the PALs were connected with educational status; a higher educational status was associated with higher PALs. Given that individuals in the community had higher PALs during the pandemic, this result can be explained as follows. Specifically, individuals’ educational status has been regularly documented as a predictor of PALs across several age groups (i.e., adolescents, adults, and the elderly) [66,67]. A prospective study that evaluated changes in individuals’ PALs during a 6-year period in people from two age groups (younger than 45 years and older than 45 years) found that less educated participants were more likely to have decreased PALs during the follow-up in both age groups [67]. We can undoubtedly support the authors’ suggestion that “participants with better educational status were probably more aware of the importance of maintaining PAL for preserving health” [67]. Additionally, previous studies have suggested that education plays a role in the self-regulation of PA, which means that better educated individuals turn intentions into behavior (i.e., PA) [68]. 

### 4.3. Limitations and Strengths

This study has several limitations. First, due to the cross-sectional nature of the study, we could not undoubtedly determine the cause–effect relationships between variables. In general, while PALs could be influenced by some of the studied factors, there is also a certain possibility that the PALs themselves were the cause of the status of some of the observed variables. Next, the validation of the PASE was performed a few months later than the main part of the study, but this was due to specific conditions and the fact that contact with participants was limited during the main part of the investigation (pandemic period). Finally, as in any other interview-based investigation, the honesty of the responses can be questioned. However, considering that we did not obtain answers to problematic questions, we believe that the collected data represent the status of the participants. 

To the best of our knowledge, this is one of the first studies in the region to investigate the correlates of PALs in elderly individuals during the pandemic period. Also, the fact that we were able to differentially evaluate the factors associated with PALs in community-dwelling and nursing home residents is an important strength of this study. Therefore, although we are aware that these results are not the final word on this topic, we hope that they will allow for further and more profound research on this problem. 

## 5. Conclusions

The translated version of the PASE was found to be a reliable and valid tool in the evaluation of PALs among elderly individuals from Croatia and Bosnia and Herzegovina. Considering the similarity of the languages spoken in the territory, we may suggest its usage in other surrounding countries (e.g., other territories of former Yugoslavia). However, before its application, the translated version of the PASE provided here should be eventually modified and adapted specifically to the characteristics of the sample (i.e., modifying the types of activities and sports). 

Once again, individuals’ health status and age were found to be the most important correlates of PALs in the elderly. In general, higher PALs are strongly negatively associated with age, and positively associated with a better health status. Therefore, we suggest that future studies evaluating the factors associated with PALs in this age group include participants’ age and health status as important (non-modifiable) confounding factors in analyses. 

When observing the total sample of participants, higher pandemic PALs were associated with community-dwelling residents. It is likely that the COVID-19-imposed measures of social distancing at least partially influenced these results. Therefore, future studies should evaluate the established association during the non-pandemic period in order to better understand the problem, and to provide adequate measures for improving PALs in the elderly population. 

The results highlighted the specific correlates of PALs among community-dwelling and nursing home residents in the pandemic period. While pre-pandemic PALs were positively correlated with pandemic PALs in nursing home residents, education status was positively correlated with pandemic PALs among community-dwelling residents. Once again, it is likely that the specific associations we evidenced here are a consequence of the period in which the study was performed. 

Finally, we can draw some most important implications of the study which would hopefully help in eventual similar situations. First, special attention should be placed on older participants of lower educational level. Most importantly, they should be informed on the importance and benefits of PA. Second, the elderly should be instructed about the most appropriate ways of physical exercising and the consequent increase of the PAL in situations when regular activities are limited. This altogether highlights the importance of physical literacy in reaching the appropriate PALs in the elderly. 

## Figures and Tables

**Figure 1 behavsci-14-00062-f001:**
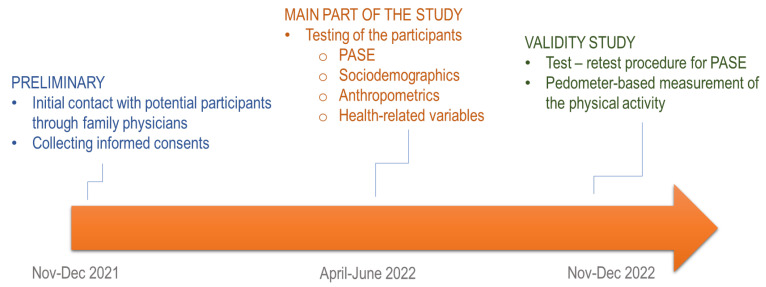
Study design, time-frames, and testing.

**Figure 2 behavsci-14-00062-f002:**
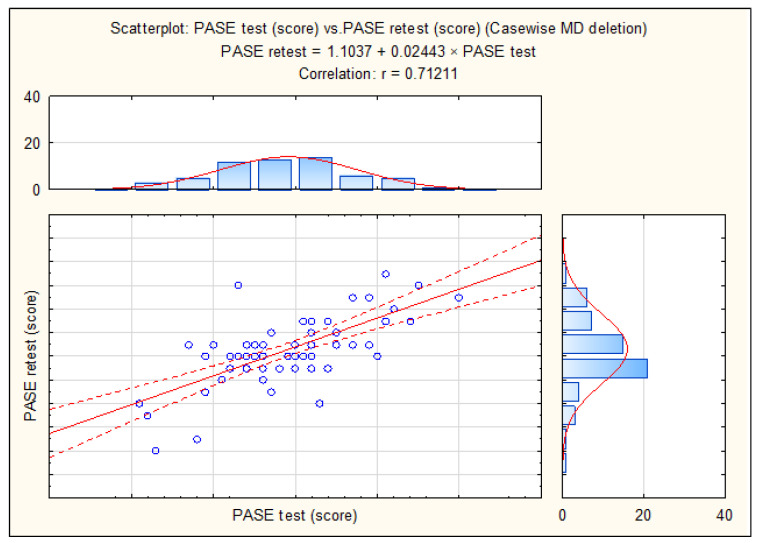
Test–retest correlation of the Physical Activity Scale for the Elderly (PASE).

**Figure 3 behavsci-14-00062-f003:**
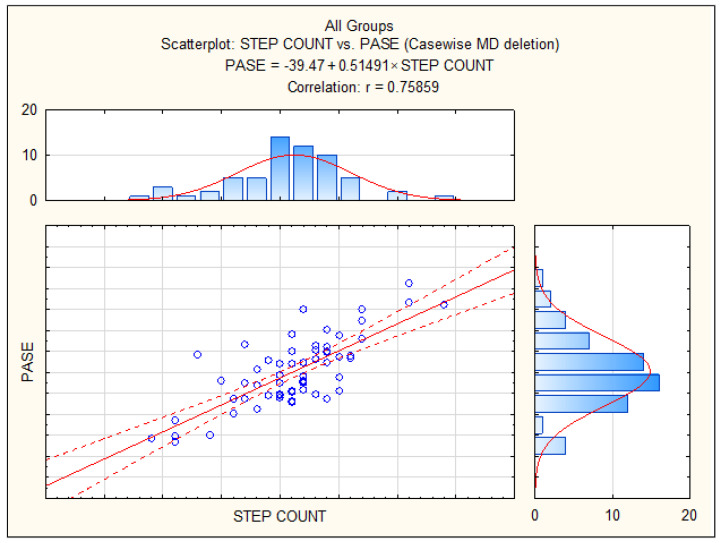
Correlations between objectively measured physical activity level (number of steps per day) and Physical Activity Scale for the Elderly (PASE) for the total sample.

**Figure 4 behavsci-14-00062-f004:**
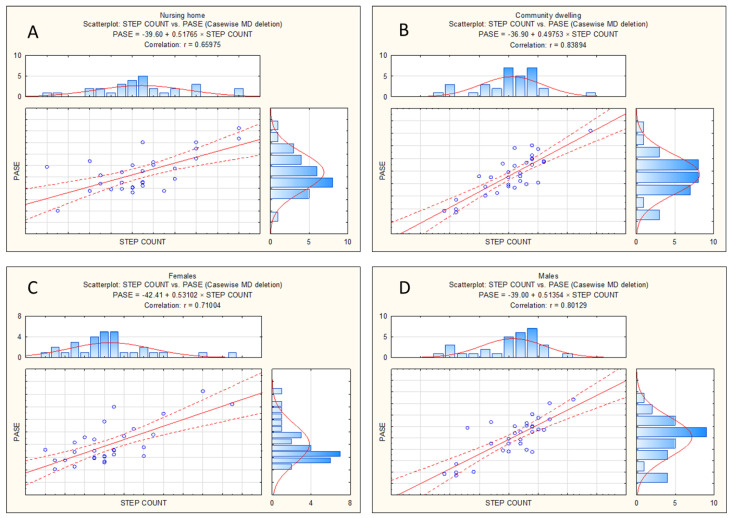
Correlations between objectively measured physical activity levels (number of steps per day) and Physical Activity Scale for the Elderly (PASE) for nursing home residents (**A**), community-dwelling residents (**B**), females (**C**), and males (**D**).

**Figure 5 behavsci-14-00062-f005:**
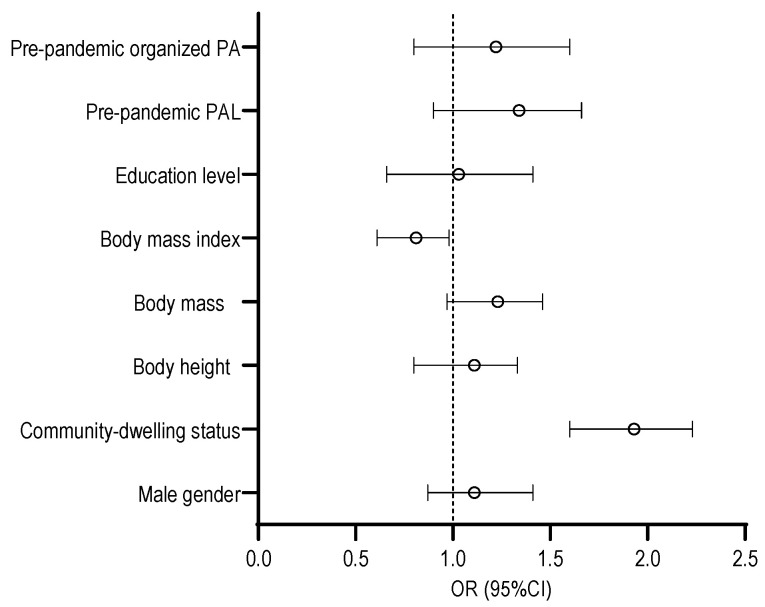
Logistic regression results with study variables as predictors and physical activity obtained by Physical Activity Scale for the Elderly (PASE) for the total sample of participants.

**Figure 6 behavsci-14-00062-f006:**
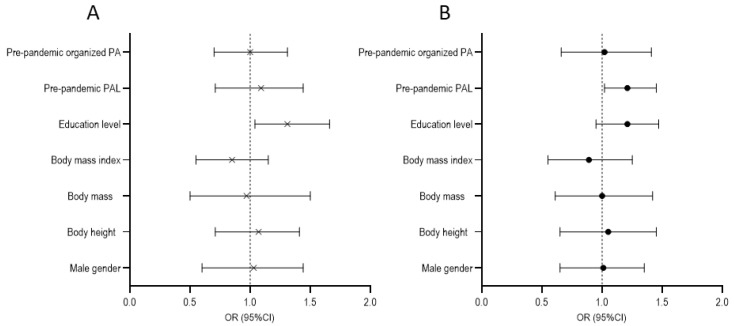
Logistic regression results with study variables as predictors and physical activity obtained by Physical Activity Scale for the Elderly (PASE) for community-dwelling residents (**A**) and nursing home residents (**B**).

**Table 1 behavsci-14-00062-t001:** Descriptive statistics (data are presented as means ± standard deviations) and gender differences in the studied variables calculated using a *t*-test for independent samples.

	Total Sample (*n* = 211)	Males (*n* = 110)	Females (*n* = 101)
Age (years)	71.44 ± 3.54	71.12 ± 4.11	72.01 ± 4.54
Body height (cm)	163.11 ± 4.58	166.11 ± 4.01	161.21 ± 3.11 *
Body mass (kg)	73.13 ± 6.01	74.13 ± 3.22	72.41 ± 5.11 *
Body mass index (kg/m^2^)	27.21 ± 5.01	26.65 ± 3.33	28.2 ± 4.00 *
SF-12_MCS (score)	55.02 ± 10.11	54.12 ± 8.33	56.81 ± 11.12
SF-12_PCS (score)	49.44 ± 8.31	50.56 ± 10.13	48.91 ± 7.84
PASE (score)	156.32 ± 40.12	161.32 ± 39.18	152.31 ± 40.72

Legend: SF-12_MCS Short Form Health Survey mental capacity score, SF-12_ PCS—Short Form Health Survey physical capacity score, * denotes statistically significant differences between genders at *p* < 0.05.

## Data Availability

Data availability is limited due to contract with Ministry. Data are all available to all interested parties upon reasonable request.

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
