# Peer review of "Exploring Factors Associated with Physical Activity in the Elderly: A Cross-Sectional Study during the COVID-19 Pandemic"

_behavsci, 2024, doi:10.3390/bs14010062_

Round 1
Reviewer 1 Report
Comments and Suggestions for Authors
Dear authors,
Thank you for the opportunity to read your manuscript. Your work examining the physical activity levels of the elderly in your region provides interesting insight into those subpopulations that may require additional support as we recover from the pandemic. I have a few suggestions for your manuscript that I have included below to clarify and expand upon some of the points mentioned. Wishing you all the best with your revisions!
Abstract
Lines 24-25: Age is mentioned twice in the sentence.
Introduction
One of the main aspects of the study is housing, but this is not mentioned in the introduction. Add in a reasoning for how dwellings can influence health behaviours.
It would be interesting to add some background on why there needs to be more focus on the Southeastern Europe population. Are there unique lifestyle differences, social norms, geographies, etc. that make that make the area unique that should be highlighted?
Methods
Lines 107-109: can you be more specific on what the regulations were? This will help provide other counties with context if the findings can potentially be generalizable to them.
Lines 160-161: Can you add more context into how you conducted your interviews? What steps did you have in place to ensure consistent interviews when there are multiple interviewers? Did you just ask the PASE-related questions or did you have probes as well to get more context?
Results
Line 248: It would be beneficial visually to have subheadings to break up the findings from your validity study and the PAL correlates evaluation.
Line 272 and 274: are you sure you mean Figure 3? Do you mean Figure 6 A and B?
Line 279: do you mean 6A and B?
Discussion
Line 282: Make sure to add that you are specifically looking at the correlates of PAL for the elderly.
Line 319: ‘Both issues’ doesn’t make sense in this context as you have only listed one issue in the previous sentence.
Lines 334-33: to make the paper accessible to those who don’t work or conduct research on the elderly, be clear on why we will see those differences between community dwellings versus nursing homes. Arguably, having organized activities can facilitate physical activity versus those who need to be self-directed.
Line 338: Do you mean it is lower in nursing home residents?
Line 347: I would avoid writing statements like “it is relatively straightforward” as it assumes that the reader is knowledgeable in this field. Explain a brief reasoning for why we are seeing differences between age groups.
Line 351: add references to this statement.
Line 415: Another thing to consider as this was during the pandemic…due to social distancing rules, were there lower PALs in nursing homes due to the loss of regular activities? Were they at limited capacity and or not available at all?
Conclusion
Add in the implications of your findings. How can it be used in practice to support COVID-19 recovery?
Author Response
Dear authors,
Thank you for the opportunity to read your manuscript. Your work examining the physical activity levels of the elderly in your region provides interesting insight into those subpopulations that may require additional support as we recover from the pandemic. I have a few suggestions for your manuscript that I have included below to clarify and expand upon some of the points mentioned. Wishing you all the best with your revisions!
RESPONSE: Thank you for recognizing the quality of our work. Also, we are particularly grateful to your comments. We tried to follow it strictly and amended the manuscript accordingly. Please see bellow how we dealt with each of your suggestion and where to find amendments in the manuscript file. All changes are indicated by "track changes" in the manuscript file. Staying at your disposal!
Abstract
Lines 24-25: Age is mentioned twice in the sentence.
RESPONSE: Thank you for noticing it. It is corrected, and sentence reads: “The participants were 211 persons older than 65 years (101 females), of whom 111 were community-dwelling residents, and 110 were nursing home residents (71.11 ± 3.11 and 72.22 ± 4.01 years of age, respectively; t-test = 0.91, p <0.05).” Please see highlighted text in the Abstract
Introduction
One of the main aspects of the study is housing, but this is not mentioned in the introduction. Add in a reasoning for how dwellings can influence health behaviours.
RESPONSE: Indeed, we missed to introduce the problem of housing as important aspect of our study. It is now corrected, and explained. Text reads: “One of the important aspects of health status in elderly is housing [8]. This is particularly the case considering the differences in PAL and factors associated with between older people living in their own homes (e.g. community-dwelling people), and those living in nursing homes (institutions) [9,10]. Specifically, in long-term care institutions, barriers to physical activity include personal factors such as health problems and fear of injury, as well as environmental factors like lack of understanding and restrictions [11,12]. On the other hand, among older people living at home, customary physical activity is generally low, with age, health status and sex being key determinants [13].” (please see 2nd paragraph of the Introduction).
It would be interesting to add some background on why there needs to be more focus on the Southeastern Europe population. Are there unique lifestyle differences, social norms, geographies, etc. that make that make the area unique that should be highlighted?
RESPONSE: Thank you for this suggestion. Indeed, this region is specific and we tried to briefly explain it in this version of the manuscript. Text reads: “Considering the specifics of this region, the investigation of the PAL in elderly would be particularly important. In brief, (older) people living in this region experienced wars in the early 1990’s, which dramatically changed their lives, and in some cases places of residence (due to large migrations of the ethnic groups in the territory of former Yugoslavia). Also, over the last 10 years EU member countries are facing economic migrations in west-ern EU countries, which logically result in a lack of familial support for elderly people [29].” (please see last paragraph of the Introduction).
Methods
Lines 107-109: can you be more specific on what the regulations were? This will help provide other counties with context if the findings can potentially be generalizable to them.
RESPONSE: Thank you for this suggestion. It is explained now, and text reads: “. Specifically, masks were mandatory in all health institutions and public transport, nursing home visits were limited and visitors were health-checked, while social distancing of 1.5 meters in public places was mandatory.” (please see 1st paragraph of the Materials section)
Lines 160-161: Can you add more context into how you conducted your interviews? What steps did you have in place to ensure consistent interviews when there are multiple interviewers? Did you just ask the PASE-related questions or did you have probes as well to get more context?
RESPONSE: Thank you. We tried to present the interviewing protocol, and text reads: “The interviews were conducted by experienced interviewers, the authors of this paper. While practically all questions were standardized, we generally performed structured interviews, while only in some cases additional explanations were needed (i.e. when participant didn’t recognize the question, and interviewer had to present some type of physical activity or health condition in more figurative speech). Ordering and wording of the questions were similar for all interviewees, and pre-defined a specific protocol was used during interviews (i.e. personal presentation of the interviewer, introduction, main part of interviewing, conclusion). Such protocol minimized the potential biases and the required time, and the conductor controlled the process.” (please see 3rd paragraph of the Variables subsection)
Results
Line 248: It would be beneficial visually to have subheadings to break up the findings from your validity study and the PAL correlates evaluation.
RESPONSE: Thank you. Subheadings are added as suggested.
Line 272 and 274: are you sure you mean Figure 3? Do you mean Figure 6 A and B?
RESPONSE: By all means, thank you very much! Amended accordingly.
Line 279: do you mean 6A and B?
RESPONSE: Same as previously. Thank you, corrected!
Discussion
Line 282: Make sure to add that you are specifically looking at the correlates of PAL for the elderly.
RESPONSE: Thank you. Sentence is amended and now reads: “The aim of this research was to investigate the factors associated with PAL during the COVID-19 pandemic in elderly.” (please see first sentence of the Discussion section).
Line 319: ‘Both issues’ doesn’t make sense in this context as you have only listed one issue in the previous sentence.
RESPONSE: Thank you, it is corrected and now reads: “This is explainable with regard to the structure of the PASE itself, and differences between the types of PAL: (i) between males and females, and (ii) between participants living in nursing homes and community-dwelling residents.” (please see 3rd paragraph of the subheading 4.1. Psychometric properties)
Lines 334-33: to make the paper accessible to those who don’t work or conduct research on the elderly, be clear on why we will see those differences between community dwellings versus nursing homes. Arguably, having organized activities can facilitate physical activity versus those who need to be self-directed.
RESPONSE: Yes, off course. We tried to highlight that community dwelling participants were in situation to do more steps simply because of regular home duties (going to shops, pharmacies, etc.). This is explained now and text reads: “A similar explanation is probable even for the higher correlation between the PASE and step count in community-dwelling participants than in nursing home residents. In brief, community-dwelling participants were more likely to take a higher number of steps per day simply because of the regular home duties. They individually went to shops, pharmacies, prepared the food, did the cleaning, etc., which altogether increase the number of steps. Consequently, their PAL is directly influenced by “step count”. Meanwhile, this is not so likely for nursing home residents, whose PALs are mostly associated with organized recreation in nursing homes, and in most cases are not based on (higher) number of steps, but are based on standing-, sitting-exercises.” (please see 5th paragraph of the subsection 4.1. Psychometric properties). Thank you!
Line 338: Do you mean it is lower in nursing home residents?
RESPONSE: Thank you. We amended it accordingly.
Line 347: I would avoid writing statements like “it is relatively straightforward” as it assumes that the reader is knowledgeable in this field. Explain a brief reasoning for why we are seeing differences between age groups.
RESPONSE: Thank you for this suggestion. Accordingly, we tried to provide a brief explanation for the negative correlation between age and PAL emphasizing the previous studies in the field. Text reads: “The negative correlation between age and PALs is well documented, and explained by numerous factors including lower functional fitness, higher prevalence of depression, different psychosocial factors (i.e. lack family support, poor awareness of exercise benefits, and expectations) [44,45]. Most probably these explanations are plausible in our study as well. However, we will briefly discuss the association between health status and PALs.” (please see first paragraph of the subsection 4.2. Predictors)
Line 351: add references to this statement.
RESPONSE: Thank you, references are added and text reads: “Specifically, people with good health conditions were more likely to have the capacity to participate in any form of PA, and such a relationship is directly and indirectly confirmed in numerous studies [46,47].” (please see beginning of the 2nd paragraph of the subsection 4.2. Predictors…)
Line 415: Another thing to consider as this was during the pandemic…due to social distancing rules, were there lower PALs in nursing homes due to the loss of regular activities? Were they at limited capacity and or not available at all?
RESPONSE: Yes, this is another factor which should be considered. Actually, regular PA activities were not organized so frequently because authorities avoided exercising in the closed facilities and rooms but PA was arranged only in appropriate weather conditions. It is added as an explanation and text reads: “However, during the pandemic period, due to limitations on social gatherings, such activities in nursing homes were limited, and were organized only when weather conditions allowed exercising in open space.” (please see 6th paragraph of the subsection 4.2. Predictors…)
Conclusion
Add in the implications of your findings. How can it be used in practice to support COVID-19 recovery?
RESPONSE: Thank you, we added one paragraph of the implications in the Conclusion, and text reads: “Finally, we can draw some most important implication of the study which would hopefully help in eventual similar situations. First, special attention should be placed on older participants of lower educational level. Most importantly, they should be informed on importance and benefits of PAL. Second, elderly should be instructed about most ap-propriate ways of physical exercising and consequent increase of the PAL in situations when regular activities are limited. It altogether highlights the importance of physical literacy in reaching the appropriate PALs in the elderly.” (please see last paragraph of the Conclusion.
Staying at your disposal!
Reviewer 2 Report
Comments and Suggestions for Authors
The authors have done a great job in presenting their research.
Throughout the manuscript they give a detailed description of all the processes followed and all the results found.
In the introduction they have made a correct theoretical foundation, approaching the final object of study. The hypothesis and objectives could have been defined in a more concrete way.
The methodology includes all the necessary details.
The results are in accordance with the design, methodologies and analysis carried out.
The discussion addresses the most important aspects.
The conclusions are appropriate to the findings.
Author Response
The authors have done a great job in presenting their research.
RESPONSE: Thank you for recognizing the quality of our work and manuscript. Also, thank you for your suggestions. We tried to follow it strictly. Please see bellow for responses and amendmendts.
Throughout the manuscript they give a detailed description of all the processes followed and all the results found.
RESPONSE: Thank you. In this version of the manuscript additional details and background are added with regard to community of residence and importance of this factor in examination of correlates of physical activity. Also, some details are added with regard to specific geographical region of the study.
In the introduction they have made a correct theoretical foundation, approaching the final object of study. The hypothesis and objectives could have been defined in a more concrete way.
RESPONSE: Thank you for this suggestion. We tried to be more specific in defining study aims and hypothesis as you suggested. Therefore, study aim and hypothesis are operationalized, and now reads: “Therefore, the aim of this research was to investigate the anthropometric-, sociodemographic- and health-related-factors associated with PALs in elder participants during the pandemic period, considering their place of residence (community-dwelling and nursing home residents). Initially, we hypothesized that studied factors would be differentially associated with PALs during the pandemic period in community-dwelling and nursing home residents.” (please see last paragraph of the Introduction)
The methodology includes all the necessary details.
RESPONSE: Thank you!
The results are in accordance with the design, methodologies and analysis carried out.
RESPONSE: In this version of the manuscript the subheadings are provided in this section for better understanding.
The discussion addresses the most important aspects.
RESPONSE: Thank you. In this version some details are added in discussion, mostly related to some specific explanations of factors associated with PAL in studied participants.
Staying at your disposal!